# Morphological Characteristics of Proximal Ulna Fractures: A Proposal for a New Classification and Agreement for Validation

**DOI:** 10.3390/healthcare11050693

**Published:** 2023-02-26

**Authors:** Pedro José Labronici, William Dias Belangero, Carlos Miguel Zublin, Lucas Braga Jaques Gonçalves, Humberto Fajardo, Robinson Esteves Pires, Vincenzo Giordano

**Affiliations:** 1Departamento de Ortopedia e Traumatologia, Universidade Federal Fluminense (UFF), Av. Marquês do Paraná, 303, Niterói 24220-000, RJ, Brazil; 2Serviço de Ortopedia e Traumatologia Prof. Dr. Donato D´Ângelo, Hospital Santa Teresa, R. Paulino Afonso, 477, Petrópolis 25680-003, RJ, Brazil; 3Departmento de Ortopedia e Traumatologia, Universidade Estadual de Campinas (UNICAMP), Cidade Universitária Zeferino Vaz, Campinas 13083-970, SP, Brazil; 4Hospital de la Polícia Federal Argentina Churruca-Visca, Uspallata 3400, Buenos Aires C1437 JCP, Argentina; 5Serviço de Ortopedia, Hospital Madre Teresa, Av. Raja Gabáglia, 1002, Belo Horizonte 30441-070, MG, Brazil; 6Departamento do Aparelho Locomotor, Universidade Federal de Minas Gerais (UFMG), Serviço de Ortopedia, Hospital Felício Rocho, Av. do Contorno, 9530, Belo Horizonte 30110-934, MG, Brazil; 7Serviço de Ortopedia e Traumatologia Prof. Nova Monteiro, Hospital Municipal Miguel Couto, Rua Mário Ribeiro 117, Rio de Janeiro 22430-160, RJ, Brazil

**Keywords:** classification, proximal ulna fracture, elbow trauma, validation

## Abstract

Historically, proximal ulna fractures have been simplistically diagnosed and treated as simple olecranon fractures, leading to an unacceptable number of complications. Our hypothesis was that the recognition of lateral, intermediate, and medial stabilizers of the proximal ulna and ulnohumeral and proximal radioulnar joints would facilitate decision-making, including the choice of approach and type of fixation. The primary aim was to propose a new classification for complex fractures of the proximal ulna based on morphological characteristics seen on three-dimensional computed tomography (3D CT). The secondary aim was to validate the proposed classification regarding its intra- and inter-rater agreement. Three raters with different levels of experience analyzed 39 cases of complex fractures of the proximal ulna using radiographs and 3D CT scans. We presented the proposed classification (divided into four types with subtypes) to the raters. In this classification, the medial column of the ulna involves the sublime tubercle and is where the anterior medial collateral ligament is inserted, the lateral column contains the supinator crest and is where the lateral ulnar collateral ligament is inserted, and the intermediate column involves the coronoid process of the ulna, olecranon, and anterior capsule of the elbow. Intra- and inter-rater agreement was analyzed for two different rounds, and the results were evaluated according to Fleiss kappa, Cohen kappa, and Kendall coefficient. Intra- and inter-rater agreement values were very good (0.82 and 0.77, respectively). Good intra- and inter-rater agreement attested to the stability of the proposed classification among the raters, regardless of the level of experience of each one. The new classification proved to be easy to understand and had very good intra- and inter-rater agreement, regardless of the level of experience of each rater.

## 1. Introduction

Proximal ulna fractures account for approximately 10% of all upper limb fractures and 21% of all proximal forearm fractures [1]. Historically, these injuries have been simplistically diagnosed as simple olecranon fractures or more complex olecranon fracture–dislocations. This misinterpretation has led to an unacceptable number of complications, mainly due to poorly reduced proximal ulna fractures, resulting in limb instability, stiffness, and dysfunction [1,2,3].

A recent improvement in imaging techniques and a better understanding of the biomechanical role of the main determinants of elbow stability have been crucial to changing this situation, allowing proper identification of all morphological characteristics of the proximal ulna [4,5,6,7,8,9,10,11,12,13,14]. On the medial side, the olecranon and coronoid process act as elbow stabilizers. In particular, the coronoid process, which functions as an important primary stabilizer of this joint, has two facets, separated by a ridge that runs along the greater sigmoid notch. While the anteromedial facet acts as a primary stabilizer, the anterolateral facet is a secondary stabilizer, sharing with the radial head the valgus stabilization of the elbow. In addition to these bony structures, the anterior bundle of the medial collateral ligament, which inserts into the sublime tubercle, is another fundamental stabilizer of the elbow joint on the medial side, resisting deforming forces in varus. Laterally, the radial head acts as a secondary restrictor to valgus deformation, so that the lateral ligament complex acts statically and dynamically to restrict valgus and varus forces. In addition, the lateral ulnar collateral ligament is of paramount importance in the posterior stability of the radial head. Finally, in the sagittal plane, both the olecranon and the triceps brachialis tendon and the coronoid process and anterior capsule of the elbow act as important restrictors of anterior and posterior translation of the ulna, respectively.

Current treatment of proximal ulna fractures involves open repair or reconstruction of the osteoligamentous stabilizers of the ulnohumeral and proximal radioulnar joints, with early active mobilization postoperatively. In this context, full restoration of the anatomy of the proximal ulna and its anatomic relationships is essential, including the medial and lateral structures as well as the olecranon, coronoid process, and trochlear notch [1]. Recent literature has highlighted the importance of identifying different patterns of injury to the proximal ulna, but there is no clear guidance on the role of 360° stabilization of the elbow [15,16]. Moreover, few studies have used this concept of treatment for proximal ulna fractures [4,5,6,7,8,9,10,11,12,13,14,15,16]. Understanding the impact of injury to these osteoligamentous elements is the basis for avoiding post-traumatic elbow instability [16]. In this scenario, the elbow joint should be divided into three columns as suggested by Watts et al. [16]. These authors introduced the Wrightington classification of elbow fracture dislocation, describing the recognized injury patterns of the three columns to guide treatment decision-making. In their study [16], the radial head represents the lateral element, the anterolateral coronoid facet represents the middle element, and the anteromedial coronoid facet represents the medial element. Although these osseous structures are extremely important in the genesis of the post-traumatic instability of the dislocated elbow, other structures, such as the olecranon, supinator crest, anterior capsule of the elbow and the collateral ligaments, were not considered, which may lead to some misinterpretations in the definition of the treatment strategy for these challenging lesions.

Thus, we propose the use of a true osteoligamentous three-column concept for all proximal ulna fractures to restore 360° stability adequately and anatomically to the elbow. Our hypothesis is that recognition of the lateral, intermediate, and medial osteoligamentous stabilizers of the proximal ulna and ulnohumeral and proximal radioulnar joints that require repair will facilitate decision-making, including the choice of approach and type of internal fixation and/or ligament repair. The primary aim of our study was to propose a new classification for complex proximal ulna fractures based on morphological characteristics seen on three-dimensional computed tomography (3D CT). The secondary aim was to validate the proposed classification regarding its intra- and inter-rater agreement.

## 2. Materials and Methods

### 2.1. Patient Selection

From 2018 to 2020, 142 fractures involving the proximal ulna treated in a Brazilian hospital were analyzed retrospectively. Patients aged 18 years or over with complex transolecranon or Monteggia-like fractures of the proximal ulna were included. By definition, transolecranon fracture–dislocation preserves joint congruency, whereas Monteggia fracture–dislocation presents proximal radioulnar joint incongruency. Patients with fracture–dislocations showing posterolateral and medial instability, those without a 3D CT scan, and those with previous trauma, infection sequela, or tumor or metabolic injury to the elbow region were excluded. This study was approved by a human research ethics committee with protocol number 49409121.2.1001.5127. All patients signed an Informed Consent form.

The probability of making a Type II beta (β) error was calculated to determine the number of samples needed to detect significant changes. Thus, 39 complex fractures of the proximal ulna were randomly selected and analyzed by 3D CT. Ten (25.6%) patients were female and twenty-nine (74.4%) were male. Age ranged from 19 to 79 years with a mean of 44.4 years, a median of 43 years, and a standard deviation of 16.0 years (coefficient of variation was approximately 0.36). Twenty (51.3%) injuries were classified as Monteggia-like (Jupiter types IIA and IID) and nineteen (48.7%) as transolecranon. Of the 20 Monteggia-like injuries, 9 (45.0%) occurred in female patients and 11 (55.0%) in male patients. Of the 19 transolecranon injuries, 1 (5.3%) occurred in a female patient and 18 (94.7%) in male patients. No significant age difference was found between patients with Monteggia-like and transolecranon fractures (*p* = 0.899) (Table 1). Women were significantly older than men (*p* = 0.007, Mann–Whitney test). Figure 1 shows how the selection process used for defining the images was performed step by step.

CT scans were obtained using a SOMATOM go. Up 32-slice system (Siemens, Erlangen, Germany), and slice thickness was 0.625 mm. Axial, oblique coronal, and sagittal images adapted to the elbow plane were generated, and 3D volume-rendered images and 3D reconstruction models were obtained for all patients.

### 2.2. Classification

In the proposed classification, the medial column of the ulna involves the sublime tubercle and is where the anterior medial collateral ligament (MCL) is inserted; the lateral column of the ulna contains the supinator crest and is where the lateral ulnar collateral ligament (LUCL) is inserted; and the intermediate column involves the coronoid process of the ulna, olecranon, and anterior capsule of the elbow. Figure 2, Figure 3, Figure 4, Figure 5, Figure 6, Figure 7 and Figure 8 show the classification with the affected structures in the three columns, and the illustrated always show the medial view of the elbow on the left side and the lateral view of the elbow on the right side.

Type IA fracture—transolecranon fracture of the proximal ulna with no involvement of the sublime tubercle and supinator crest (Figure 2).

Type IB fracture—transolecranon fracture of the proximal ulna with associated fracture of the coronoid process and no involvement of the sublime tubercle and supinator crest (Figure 3).

Type IIA fracture—transolecranon fracture of the proximal ulna with associated fracture of the sublime tubercle (Figure 4).

Type IIB fracture—transolecranon fracture of the proximal ulna with associated fracture of the sublime tubercle and coronoid process (Figure 5).

Type IIIA fracture—transolecranon fracture of the proximal ulna with associated fracture of the supinator crest (Figure 6).

Type IIIB fracture—transolecranon fracture of the proximal ulna with associated fracture of the supinator crest and coronoid process (Figure 7).

Type IV fracture—transolecranon fracture of the proximal ulna with associated fracture of the sublime tubercle, supinator crest, and coronoid process (Figure 8).

Table 2 summarizes the classification of complex fractures of the proximal ulna according to the involvement of the three proposed columns.

### 2.3. Validation

All cases were assessed by three raters with different levels of experience (rater 1 [R1], 9 years of graduation in orthopedics and traumatology; rater 2 [R2], 20 years; and rater 3 [R3], 30 years). Two assessment rounds were conducted with a 30-day interval between them. Between the first and second rounds, all images were allocated and identified by the main author (P.J.L.) and the positions of cases were randomly and manually changed by the principal researcher. The raters were not allowed to keep the images after the first round, and between rounds. In both rounds, the raters had access to the classification in both descriptive and visual forms, and there was no time limit for completing the assessment.

### 2.4. Statistical Analysis

Inter-rater agreement was analyzed using Fleiss kappa weighted for agreement between the three raters and Kendall coefficient. Intra-rater agreement was analyzed using weighted Cohen kappa and Kendall coefficient. Overall analyses (full classification) and specific analyses (for each type of injury) were performed.

The kappa value represents the degree of absolute agreement between classifications. Kappa values equal to or greater than 0.75 are considered good to excellent, while values lower than 0.40 indicate poor agreement. Kendall coefficient is a measure of agreement between raters who are assessing a given set of objects. Overall, coefficient values equal to or greater than 0.90 are considered very good. A high coefficient value means that raters applied essentially the same standards to assess the samples.

## 3. Results

### 3.1. Intra-Rater Agreement

The three raters showed very good agreement between their repeated assessments. In absolute terms, R1 and R3 assigned the same classification in both assessments for 82.0% of the cases, while R2 assigned the same classification for 87% of the cases. For all raters, the significance tests attested for the significance of kappa and Kendall coefficient values (*p* < 0.05). Agreement in the two assessments by the three raters was found to be significantly different from zero, therefore very good (>0.8).

Good intra-rater agreement attested to the stability of the proposed classification between raters, regardless of the level of experience of each one. Table 3 shows the results of intra-rater agreement analysis in the two study rounds.

### 3.2. Inter-Rater Agreement

In absolute terms, inter-rater agreement was very good, as the three raters assigned the same classification for 77% of the cases in the first and second rounds. Agreement between two of the raters was greater than or equal to 0.77. All kappa and Kendall coefficient values in inter-rater analysis were greater than 0.75 (*p* < 0.001). Inter-rater agreement attested to the reproducibility of the proposed classification, regardless of the level of experience of each rater. Table 4 shows the results of inter-rater agreement analysis.

## 4. Discussion

The proposed column-based classification for fractures of the proximal ulna showed very good intra- and inter-rater agreement. Our hypothesis was that the recognition of lateral stabilizers (fractures involving the supinator crest and injury to the LUCL), intermediate stabilizers (fractures involving the coronoid process and injury to the anterior capsule), and medial stabilizers (fractures involving the sublime tubercle and the anterior band of the MCL) of the proximal ulna and ulnohumeral and proximal radioulnar joints that require repair would facilitate decision-making, including the choice of approach and type of internal fixation and/or capsuloligamentous repair. This raises new possibilities in terms of identifying the structures that require repair by 360° fixation of the elbow, especially in complex transolecranon and Monteggia-like fracture–dislocations of the proximal ulna.

Elbow joint stability can be functionally divided into static and dynamic. Static stability is controlled by the osteoarticular architecture, capsule, and ligaments, while dynamic stability is determined by neuromuscular factors. In the elbow joint, this specifically means that static stability is primarily attributed to the congruence between the joint surfaces, anterior joint capsule, and medial and lateral collateral ligaments, particularly the LUCL [17,18,19,20,21]. The intermediate column, formed by the olecranon, coronoid process, and anterior capsule of the elbow, surrounds the trochlea through an arc of approximately 170° (trochlear notch), acting as a primary osteocapsular constraint to deforming forces in the sagittal plane and as a secondary constraint in the coronal plane, especially with the elbow at maximum extension. The medial column is formed by the medial facet of the ulna and extends proximally to the base of the coronoid process and distally to the sublime tubercle; it is the site of ulnar insertion of the anterior band of the MCL and acts as a primary constraint to coronal valgus instability [22,23]. The lateral column is formed by the supinator crest, radial notch of the ulna, and LUCL, and this complex capsuloligamentous structure acts as a primary constraint to varus in the coronal plane [24,25,26,27,28]. The radial head is relatively incongruent with a greater arc of curvature than the humeral capitellum, acting as a secondary constraint to both valgus deformity in the coronal plane and posterolateral translation.

Several classifications have been proposed to describe fractures of the proximal ulna. Most of them have expanded knowledge to specific types of fracture, such as those of the olecranon and coronoid process, or associated fracture patterns, such as Monteggia and Monteggia-like injuries [29,30,31]. Although reduction and anatomic fixation of these fractures are recommended, to our knowledge, there is a lack of attention to the importance of adequate reconstruction of all lateral, intermediate, and medial osteoligamentous stabilizers of the proximal ulna and ulnohumeral and proximal radioulnar joints. Melamed et al. reviewed the results of plating of various fracture patterns of the proximal ulna, including isolated olecranon fractures, olecranon fractures combined with a coronoid fracture, and olecranon fractures combined with a coronoid and radial head fracture [32]. The authors presented a scheme that describes elbow stability in different fracture patterns and the treatment chosen for each subgroup. Although they mention the need for coronoid fixation and lateral ligament complex repair in some cases, there is no description of fixation of both the medial and lateral columns of the proximal ulna.

Unlike those authors, we included fractures involving the medial facet and sublime tubercle as well as those involving the supinator crest and proximal radioulnar joint facet. In addition to radiographic evaluation, we regularly use 3D CT to facilitate the detection of all traces of fracture and joint incongruity, which ultimately correlate with capsuloligamentous injuries that are sometimes missed at first. Midtgaard et al. demonstrated in a biomechanical study that inferior humeral translation relative to the forearm on initial lateral radiograph of the elbow suggests injury to the collateral ligaments [33]. When inferior translation exceeds 3 mm, the integrity of the lateral collateral ligament (LCL) or MCL should be questioned. If inferior translation exceeds 7.5 mm, a disruption of both ligaments (LCL and MCL) should be suspected. In the presence of a multifragmented fracture, a CT scan is necessary for preoperative planning and evaluation of the sublime tubercle and supinator crest. In the case of a capsuloligamentous injury due to a fracture at the bony insertions of the ligaments (LCL and MCL) or the anterior capsule, these fragments, even small ones, should be securely fixed. Watts et al. [16] proposed a new comprehensive classification system based on the same three-column concept of elbow fracture dislocation. These authors described a lateral, a middle, and a medial column in the elbow joint, demonstrating the natural fulcrum in between the middle and medial columns. This classification system clearly showed the impact of injury to the osseous elements for elbow instability. However, these authors did not provide any information on the ligamentous structures and their role in the genesis in both acute and chronic elbow instability.

In this context, the inclusion of all osteoligamentous structures of the elbow is probably the main strength of our study, as one can understand why minimal bone fragments can generate a major instability in this joint. Another strength of this classification for complex fractures of the proximal ulna is to help surgeons recognize areas of instability through 3D CT with a focus on the importance of the sublime tubercle, supinator crest, and coronoid process of the ulna, as well as the capsuloligamentous structures that insert into these structures. This facilitates indication for appropriate treatment based on the division of the proximal ulna into three columns (lateral, intermediate, and medial) for restoring 360° stability to the elbow. Other authors have shown that the column-based division is useful for the interpretation and management of different skeletal trauma conditions [16,34,35,36,37,38].

This study has some limitations. First, the new classification system was assessed by three orthopedic surgeons with different levels of expertise, which may have influenced our findings. However, we found good intra- and inter-rater agreement, which can be seen as a promising perspective for understanding and applying the proposed classification. Second, only transolecranon or Monteggia-like fractures of the proximal ulna were included, therefore one can ask about its validity in other complex injuries around the elbow involving the proximal region of the ulna, such as fracture–dislocations with posterolateral and/or medial instability. Although fractures of the proximal ulna vary in severity and in the diversity of associated injuries, all are characterized by presenting some degree of damage to the key stabilizing structures of the elbow [18,39,40]. The treatment principle is based on adequate reconstruction of the ulna and elbow stabilizing structures, which have been proven to be factors related to better joint function and reduced risk of progressive joint degeneration due to chronic elbow instability [39,40]. Thus, it seems logical that the 360° understanding of the elbow should be a rule and the proposed classification proved to be useful for this. Finally, we did not validate the system in a clinical setting, including a report of the outcomes of our cases treated according to the three columns of the proximal ulna. The next step for this consortium of investigators is to carry out a multicenter prospective cohort study soon to validate the new classification in a clinical–therapeutic setting.

## 5. Conclusions

The new classification for complex fractures of the proximal ulna was easily understood and can be considered reproducible, as it showed high intra- and inter-rater agreement in orthopedic surgeons with different levels of experience.

## Figures and Tables

**Figure 1 healthcare-11-00693-f001:**
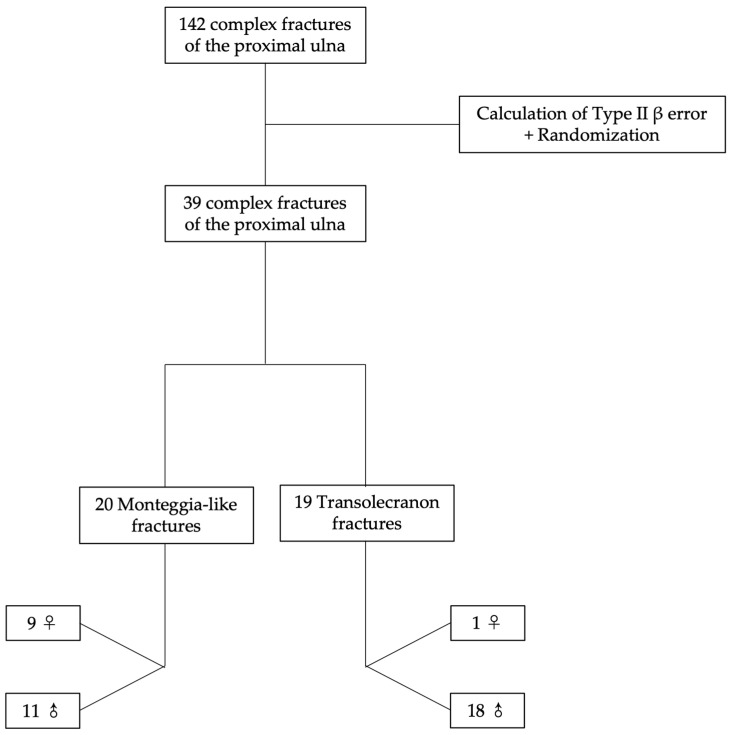
Selection process used for defining the images.

**Figure 2 healthcare-11-00693-f002:**
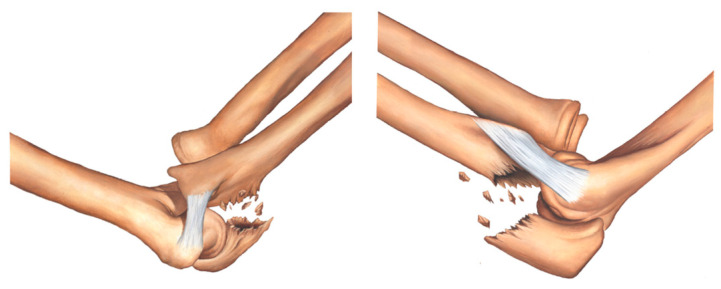
Type IA fracture—transolecranon fracture of the proximal ulna with no involvement of the sublime tubercle and supinator crest.

**Figure 3 healthcare-11-00693-f003:**
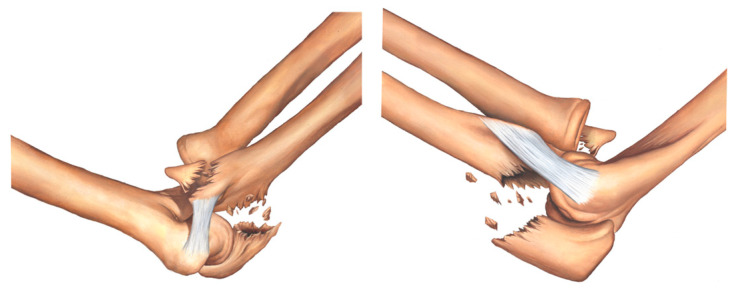
Type IB fracture—transolecranon fracture of the proximal ulna with associated fracture of the coronoid process and no involvement of the sublime tubercle and supinator crest.

**Figure 4 healthcare-11-00693-f004:**
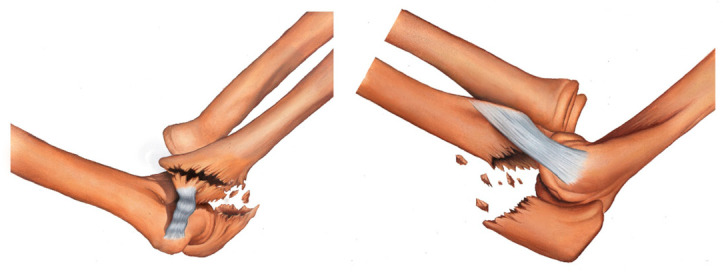
Type IIA fracture—transolecranon fracture of the proximal ulna with associated fracture of the sublime tubercle.

**Figure 5 healthcare-11-00693-f005:**
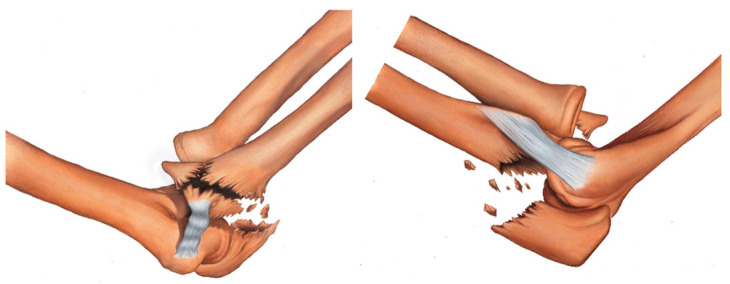
Type IIB fracture—transolecranon fracture of the proximal ulna with associated fracture of the sublime tubercle and coronoid process.

**Figure 6 healthcare-11-00693-f006:**
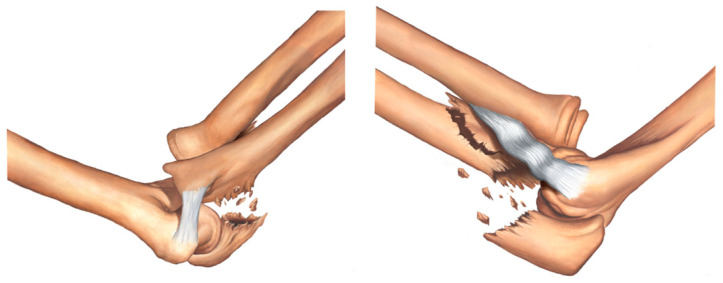
Type IIIA fracture—transolecranon fracture of the proximal ulna with associated fracture of the supinator crest.

**Figure 7 healthcare-11-00693-f007:**
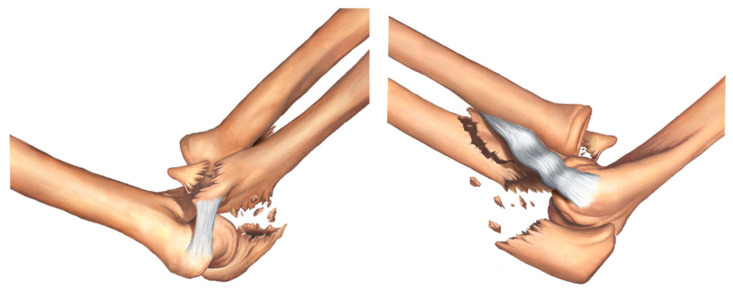
Type IIIB fracture—transolecranon fracture of the proximal ulna with associated fracture of the supinator crest and coronoid process.

**Figure 8 healthcare-11-00693-f008:**
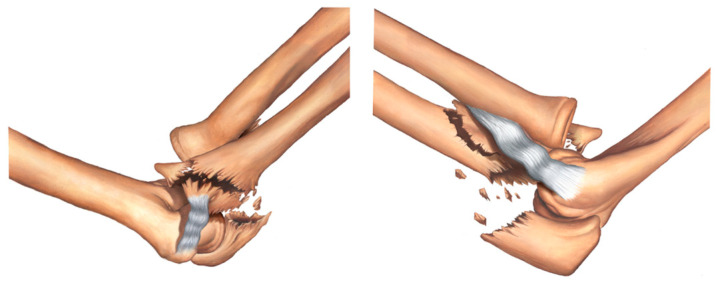
Type IV fracture—transolecranon fracture of the proximal ulna with associated fracture of the sublime tubercle, supinator crest, and coronoid process.

**Table 1 healthcare-11-00693-t001:** Age distribution for specific subgroups.

Age Statistics	Monteggia-like			Transolecranon		
	Overall	Female	Male	Overall	Female	Male
Minimum	22.0	30.0	22.0	19.0	62.0	19.0
Maximum	79.0	79.0	58.0	73.0	62.0	73.0
Median	48.0	55.0	31.0	40.0	62.0	40.0
Mean	45.2	55.7	36.5	43.7	62.0	42.7
Standard deviation	16.1	14.1	12.3	16.3	0.0	16.1
Coefficient of variation	0.36	0.25	0.34	0.37	0.00	0.38

Source: Serviço de Ortopedia e Traumatologia Prof. Donato D’Ângelo, Hospital Santa Teresa.

**Table 2 healthcare-11-00693-t002:** Affected column structures according to type in the new classification.

Type	Medial Column (Sublime Tubercle/Anterior Band of MCL)	Intermediate Column (Coronoid Process/Anterior Capsule)	Lateral Column (Supinator Crest/LUCL)
IA	−	−	−
IB	−	+	−
IIA	+	−	−
IIB	+	+	−
IIIA	−	−	+
IIIB	−	+	+
IV	+	+	+

Source: Serviço de Ortopedia e Traumatologia Prof. Donato D’Ângelo, Hospital Santa Teresa.

**Table 3 healthcare-11-00693-t003:** Analysis of intra-rater agreement in the two study rounds.

Rater	Absolute Agreement	Weighted Kappa (95% CI)	Kendall Coefficient
R1	0.82	0.82 (0.69; 0.90) **^‡^**	0.82 **^‡^**
R2	0.87	0.91 (0.83; 0.95) **^‡^**	0.84 **^‡^**
R3	0.82	0.92 (0.85; 0.96) **^‡^**	0.90 **^‡^**

Source: Serviço de Ortopedia e Traumatologia Prof. Donato D’Ângelo, Hospital Santa Teresa. **^‡^**
*p* < 0.001. CI, confidence interval; R, rater.

**Table 4 healthcare-11-00693-t004:** Analysis of inter-rater agreement in the two study rounds.

Assessment Round	Agreement Measure	R1–R2	R1–R3	R2–R3	R1, R2, and R3
1	Absolute agreement	0.79	0.77	0.77	0.77
	Weighted kappa ^‡^	0.75 (0.45; 0.85)	0.75 (0.44; 0.82)	0.78 (0.49; 0.83)	0.76 (0.59; 0.86)
	Kendall coefficient ^‡^	0.78	0.77	0.72	-
2	Absolute agreement	0.82	0.77	0.79	0.77
	Weighted kappa ^‡^	0.77 (057; 0.88)	0.75 (0.33; 0.81)	0.77 (0.44; 0.87)	0.78 (0.63; 0.88)
	Kendall coefficient ^‡^	0.78	0.77	0.78	-

Source: Serviço de Ortopedia e Traumatologia Prof. Donato D’Ângelo, Hospital Santa Teresa. **^‡^**
*p* < 0.001. R, rater.

## Data Availability

Not applicable.

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
