# Peer review of "Morphological Characteristics of Proximal Ulna Fractures: A Proposal for a New Classification and Agreement for Validation"

_healthcare, 2023, doi:10.3390/healthcare11050693_

Round 1

Reviewer 1 Report

The manuscript entitled " Morphological Characteristics of Proximal Ulna Fractures: A Proposal for a New Classification and Agreement for Validation " is of some interest. This work a presented new classification for complex fractures of the proximal ulna based on morphological characteristics seen on three-dimensional computed tomography (3D CT). The new classification for complex fractures of the proximal ulna was easily under- stood and can be considered reproducible, as it showed high intra- and inter-rater agree-ment in orthopedic surgeons with different levels of experience.However, there are several issues must be addressed as following:

1、The introduction does a decent job at presenting previous work, and presents why this approach is needed, but fails short of justifying the novelty.

2、  Is this the first report of this method? If yes, state that, if not, what other similar works have been performed and how is yours different?

3、  More details about the method should be provided, a detailed scheme figure is needed.

4、This work need to be validated in a clinical setting, Please clarify the specific research plan for the next step

Author Response

Dear Madam/Sir,

Thanks for your kind comments, all were considered and done.

Regarding comments #1 and #2, we totally agree with you. The novelty of our proposed classification was justified in the text, calling attention to the fact that, as far as we know, no other classification using the concept of three columns of the elbow brings together the osteoligamentous structures.
We believe that by mentioning the study of Watts et al. [Watts AC, Singh J, Elvey M, Hamoodi Z. Current concepts in elbow fracture dislocation. Shoulder & Elbow 2021, 13, 451-458], the difference between our study and others, including the one just mentioned is clear.
MODIFICATIONS IN THE TEXT:
Introduction, 3rd paragraph
“Recent literature has highlighted the importance of identifying different patterns of injury to the proximal ulna, but there is no clear guidance on the role of 360º stabilization of the elbow [15,16]. Also, few studies have used this concept of treatment for proximal ulna fractures [16]. Understanding the impact of injury to these osteoligamentous elements is the basis for avoiding post-traumatic elbow instability [16]. In this scenario, the elbow joint should be divided into three columns as suggested by Watts et al. [16]. These authors introduced the Wrightington classification of elbow fracture dislocation, describing the recognized the injury patterns of the three columns to guide treatment decision-making. In their study [16], the radial head represents the lateral element, the anterolateral coronoid facet represents the middle element, and the anteromedial coronoid facet represents the medial element. Although these osseous structures are extremely important in the genesis of the post-traumatic instability of the dislocated elbow, other structures, such as the olecranon, supinator crest, anterior capsule of the elbow and the collateral ligaments were not considered, which may lead to some misinterpretations in the definition of the treatment strategy for these challenging lesions.
Thus, we propose the use of a true osteoligamentous three-column concept for all proximal ulna fractures to restore 360º stability adequately and anatomically to the elbow.”

Regarding comment#3, a figure showing the selection process step by step was added and it was added in the main text “Figure 1 shows how the selection process used for defining the images was done step by step.”. ["Materials and Methods, Patient selection, 2nd paragraph, lines 108 -109]

Finally, regarding comment #4, indeed, the next step is to validate the classification in the clinical set in a prospective multicenter cohort study. This information was added as a limitation of the current study, with this caveat. Again, a new sentence was added to the text as follows “The next step for this consortium of investigators is to carry out a multicenter prospective cohort study soon to validate the new classification in a clinical-therapeutic setting.”. [Discussion, 6th paragraph, lines 317-319]

Numbered Reviewer Remark and Manuscript Line Number

Author Response

Revised Manuscript Line Number and Text Change

Reviewer 1:

The manuscript entitled " Morphological Characteristics of Proximal Ulna Fractures: A Proposal for a New Classification and Agreement for Validation " is of some interest. This work a presented new classification for complex fractures of the proximal ulna based on morphological characteristics seen on three-dimensional computed tomography (3D CT). The new classification for complex fractures of the proximal ulna was easily under- stood and can be considered reproducible, as it showed high intra- and inter-rater agreement in orthopedic surgeons with different levels of experience. However, there are several issues must be addressed as following:

Thanks for your kind comments, all were considered and done.

1.           
The introduction does a decent job at presenting previous work, and presents why this approach is needed, but fails short of justifying the novelty.

2.

Is this the first report of this method? If yes, state that, if not, what other similar works have been performed and how is yours different?

Thanks. We totally agree with you. The novelty of our proposed classification was justified in the text, calling attention to the fact that, as far as we know, no other classification using the concept of three columns of the elbow brings together the osteoligamentous structures.

We believe that by mentioning the study of Watts et al. [Watts AC, Singh J, Elvey M, Hamoodi Z. Current concepts in elbow fracture dislocation. Shoulder & Elbow 2021, 13, 451-458], the difference between our study and others, including the one just mentioned is clear.

Introduction, 3rd paragraph

“Recent literature has highlighted the importance of identifying different patterns of injury to the proximal ulna, but there is no clear guidance on the role of 360º stabilization of the elbow [15,16]. Also, few studies have used this concept of treatment for proximal ulna fractures [16]. Understanding the impact of injury to these osteoligamentous elements is the basis for avoiding post-traumatic elbow instability [16]. In this scenario, the elbow joint should be divided into three columns as suggested by Watts et al. [16]. These authors introduced the Wrightington classification of elbow fracture dislocation, describing the recognized the injury patterns of the three columns to guide treatment decision-making. In their study [16], the radial head represents the lateral element, the anterolateral coronoid facet represents the middle element, and the anteromedial coronoid facet represents the medial element. Although these osseous structures are extremely important in the genesis of the post-traumatic instability of the dislocated elbow, other structures, such as the olecranon, supinator crest, anterior capsule of the elbow and the collateral ligaments were not considered, which may lead to some misinterpretations in the definition of the treatment strategy for these challenging lesions.

Thus, we propose the use of a true osteoligamentous three-column concept for all proximal ulna fractures to restore 360º stability adequately and anatomically to the elbow.”

3.

More details about the method should be provided, a detailed scheme figure is needed.

Thanks for the suggestion. A figure showing the selection process step by step was added.

Materials and Methods, Patient selection, 2nd paragraph, lines 108 -109

“Figure 1 shows how the selection process used for defining the images was done step by step.”

4.

This work needs to be validated in a clinical setting, Please clarify the specific research plan for the next step

Indeed. The next step is to validate the classification in the clinical set in a prospective multicenter cohort study. This information was added as a limitation of the current study, with this caveat.

Discussion, 6th paragraph, lines 317-319

“The next step for this consortium of investigators is to carry out a multicenter prospective cohort study soon to validate the new classification in a clinical-therapeutic setting.”

For more details please see the revised version manuscript.

Reviewer 2 Report

The work is devoted to an actual and not completely solved problem of orthopedics - injuries of the elbow joint: fractures of the proximal ulna. There remains a fairly high percentage of unsatisfactory functional results in the treatment of this pathology: joint contractures, instability and early development of post-traumatic osteoarthritis. This is due to the persisting mechanistic approach to the choice of surgical technique without taking into account the nature of bone pathology and damage to the capsular-ligamentous apparatus.

The proposed classification of fractures of the proximal ulna is novel, because it uses the concept of columns in assessing bone damage to the ulna and osteoligamentous stabilizers. Based on the use of radiography, 3D-CT, simple, accessible and understandable. It allows to give a full assessment of the nature of the damage and to conduct a full-fledged preoperative planning in patients with transolecranon and Monteggia-like fractures with the possibility of expert evaluation of long-term results of treatment.

The material used (39 patients) and the statistical evaluation performed: general analysis and specific analysis allow us to conclude that the proposed classification is objective and valid.

For the final assessment of the proposed classification, it is necessary to recommend test it in the hospital settings, followed by an analysis of the obtained treatment results to assess the prospects for applying the proposed classification in clinics.

Author Response

Dear Madam/Sir,

Thanks for the comment. We totally agree that the next step is the clinical evaluation. As a matter of fact, the next step is to validate the classification in the clinical set in a prospective multicenter cohort study. This information was added as a limitation of the current study, with this caveat.

MODIFICATION IN THE TEXT:

“The next step for this consortium of investigators is to carry out a multicenter prospective cohort study soon to validate the new classification in a clinical-therapeutic setting.”. [Discussion, 6th paragraph, lines 317-319]

Numbered Reviewer Remark and Manuscript Line Number

Author Response

Revised Manuscript Line Number and Text Change

Reviewer 2:

1.           
The work is devoted to an actual and not completely solved problem of orthopedics - injuries of the elbow joint: fractures of the proximal ulna. There remains a fairly high percentage of unsatisfactory functional results in the treatment of this pathology: joint contractures, instability and early development of post-traumatic osteoarthritis. This is due to the persisting mechanistic approach to the choice of surgical technique without taking into account the nature of bone pathology and damage to the capsular-ligamentous apparatus.

The proposed classification of fractures of the proximal ulna is novel, because it uses the concept of columns in assessing bone damage to the ulna and osteoligamentous stabilizers. Based on the use of radiography, 3D-CT, simple, accessible and understandable. It allows to give a full assessment of the nature of the damage and to conduct a full-fledged preoperative planning in patients with transolecranon and Monteggia-like fractures with the possibility of expert evaluation of long-term results of treatment.

The material used (39 patients) and the statistical evaluation performed: general analysis and specific analysis allow us to conclude that the proposed classification is objective and valid.

For the final assessment of the proposed classification, it is necessary to recommend test it in the hospital settings, followed by an analysis of the obtained treatment results to assess the prospects for applying the proposed classification in clinics.

Thanks for the comment. We totally agree that the next step is the clinical evaluation. As a matter of fact, the next step is to validate the classification in the clinical set in a prospective multicenter cohort study. This information was added as a limitation of the current study, with this caveat.

Discussion, 6th paragraph, lines 317-319

“The next step for this consortium of investigators is to carry out a multicenter prospective cohort study soon to validate the new classification in a clinical-therapeutic setting.”

For more details please see the revised version manuscript.

Reviewer 3 Report

The submitted manuscript presents an alternative classification for ulna fractures based on computed tomography and radiographic analysis. The procedure was validated using its intra- and inter-rater agreement using the three rates with different expertise profiles and a random process to eliminate mistakes in the validation.

This work and how it’s been addressed match the Healthcare Journal’s aim and scope. In addition, the information and how it is organized are well done and could interest the medical and surgeon communities.

Although the manuscript is well organized and written, and the information presented is significant to the field, some mistakes should be taken into account:

Introduction

  1. There needs to be more background information regarding what the referenced authors have found, which could support how this work can improve what already exists.

Materials and methods

2.1  Patient selection

2.     Is there informed consent from the patients for the use of their data, or is this established in the protocol approved by the ethics committee? It is necessary to make it clear or show that there is evidence of this.

  1. Is age and gender classification necessary? Is this included in the analysis?

4.     The methodology for obtaining the tomography (CT) is mentioned; however, the methodology used to analyze the data to obtain the information needs to be clarified. It is necessary to include it.

2.2 Classification

5.     Are the images in this section based on some reference, or are they your own? Because there is no support.

6.     It could be seen better and easier to read if the classification with the images is organized as columns or in a table (As a recommendation). The view must also be identified in each image of each figure

2.3 Validation

7.     How to guarantee that the position of the cases changes randomly throughout the process and does not repeat itself simultaneously?

Results

3.1 Intra-rater agreement

8.     Although a good agreement is shown between the 3 raters involved in the test, in order to carry out a validation process regarding this type of fracture, more opinions are required to allow a complete statistical approximation.

Discussion

9.     The conclusion of a "very good" agreement can be risky, as mentioned. It should be kept clear that only 3 rates were considered.

10.  In general, the errors that could be associated with a lousy classification following this methodology, what implications do they have, or could they have for the patient?

Author Response

Dear Madam/Sir,

Thanks for your kind comments, all were considered and done. We appreciate and hope that we have been able to respond adequately all of them.

Below we'll answer all your comments:

- Regarding your first comment (Introduction), as a matter of fact, a thorough explanation about the function of the osteoligamentous structures of the elbow was added, as well as the novelty of our proposed classification was justified in the text. As far as we know, no other classification using the concept of three columns of the elbow brings together the osteoligamentous structures, therefore the aim of our study was to propose and validate the new three-column classification.

- About Material and Methods, Patient selection, regarding your first question, all patients signed an Informed Consent form. This information was added in the text. Regarding your second question about age and gender classification, this is mentioned in the 2nd paragraph. As highlighted in the end of this paragraph, women were significantly older than men (p=0.007, Mann-Whitney test). Finally, regarding your third question, using the axial, oblique coronal, and sagittal images, and 3D reconstruction, the proposed classification was applied using as markers the sublime tubercle as the medial column; the supinator crest as the lateral column; and the coronoid process of the ulna and the olecranon as the intermediate column of the ulna. As this is explained during the Material and Methods section, as you agree, we feel this don’t need further detailing.

- About Materials and methods, Classification, responding your first question, all images are from the authors’ personal archive. The source is not mentioned, since the classification is nine and proposed in the study, and because it does not belong to any other author. In regards of your second question, Table 2 shows affected column structures according to type in the new classification. The view was identified in each figure. Thanks for the suggestion.

- About Materials and methods, Validation, thanks for mentioning this aspect. The information was added in this section.

- About  Results, Intra-rater agreement, the validation criterion of the classification proposed in this study was not in the clinical setting. In this first phase of the classification development process, based on the theory of the 3 osteoligamentous columns of the elbow, the types were generated, and their validity was assessed to understand their degree of understanding and intra- and interobserver agreement. Our study showed good intra- and inter-rater agreement, attesting to the stability of the proposed classification among the raters, regardless of the level of experience of each one. The next phase of this study is to assess its usefulness in the clinical scenario, as stated in lines 317 to 319 of the Discussion.

- About Discussion, Thanks again. Regarding your first comment, this limitation is clearly stated in the last paragraph of the Discussion section, where we write “First, the new classification system was assessed by three orthopedic surgeons with dif-ferent levels of expertise, which may have influenced our findings. However, we found good intra- and inter-rater agreement, which can be seen as a promising perspective for understanding and applying the proposed classification.”. Finally, regarding the errors that may be associated with a poor classification according to the methodology used in the study, it is not possible to know or even understand the potential implications they have for the patient. As mentioned, this is phase 1 of the complete validation of our classification. However, the concept of three columns for the elbow joint is not new. The novelty of our classification is to gather all the osteoligamentous structures of the medial, intermediate (or middle) and lateral columns. Theoretically, there are no major implications, as we are using anatomical landmarks and functional structures for stabilization and normal range of motion of this joint.

Numbered Reviewer Remark and Manuscript Line Number

Author Response

Revised Manuscript Line Number and Text Change

Reviewer 3:

The submitted manuscript presents an alternative classification for ulna fractures based on computed tomography and radiographic analysis. The procedure was validated using its intra- and inter-rater agreement using the three rates with different expertise profiles and a random process to eliminate mistakes in the validation.

This work and how it’s been addressed match the Healthcare Journal’s aim and scope. In addition, the information and how it is organized are well done and could interest the medical and surgeon communities.

Although the manuscript is well organized and written, and the information presented is significant to the field, some mistakes should be taken into account:

Thanks for your kind comments, all were considered and done. We appreciate and hope that we have been able to respond adequately all of them.

1.

Introduction

There needs to be more background information regarding what the referenced authors have found, which could support how this work can improve what already exists.

Thanks. We fully agree. As a matter of fact, a thorough explanation about the function of the osteoligamentous structures of the elbow was added, as well as the novelty of our proposed classification was justified in the text. As far as we know, no other classification using the concept of three columns of the elbow brings together the osteoligamentous structures, therefore the aim of our study was to propose and validate the new three-column classification.

Introduction, 2nd paragraph, lines 56-69

“On the medial side, the olecranon and coronoid process act as elbow stabilizers. In particular, the coronoid process, which functions as an important primary stabilizer of this joint, has two facets, separated by a ridge that runs along the greater sigmoid notch. While the anteromedial facet acts as a primary stabilizer, the anterolateral facet is a secondary stabilizer, sharing with the radial head the valgus stabilization of the elbow. In addition to these bony structures, the anterior bundle of the medial collateral ligament, which inserts into the sublime tubercle, is another fundamental stabilizer of the elbow joint on the medial side, resisting deforming forces in varus. Laterally, the radial head acts as a secondary restrictor to valgus deformation, so that the lateral ligament complex acts statically and dynamically to restrict valgus and varus forces. In addition, the lateral ulnar collateral ligament is of paramount importance in the posterior stability of the radial head. Finally, in the sagittal plane, both the olecranon and the triceps brachialis tendon and the coronoid process and anterior capsule of the elbow act as important restrictors of anterior and posterior translation of the ulna, respectively.”

Introduction, 3rd paragraph

“Recent literature has highlighted the importance of identifying different patterns of injury to the proximal ulna, but there is no clear guidance on the role of 360º stabilization of the elbow [15,16]. Also, few studies have used this concept of treatment for proximal ulna fractures [16]. Understanding the impact of injury to these osteoligamentous elements is the basis for avoiding post-traumatic elbow instability [16]. In this scenario, the elbow joint should be divided into three columns as suggested by Watts et al. [16]. These authors introduced the Wrightington classification of elbow fracture dislocation, describing the recognized the injury patterns of the three columns to guide treatment decision-making. In their study [16], the radial head represents the lateral element, the anterolateral coronoid facet represents the middle element, and the anteromedial coronoid facet represents the medial element. Although these osseous structures are extremely important in the genesis of the post-traumatic instability of the dislocated elbow, other structures, such as the olecranon, supinator crest, anterior capsule of the elbow and the collateral ligaments were not considered, which may lead to some misinterpretations in the definition of the treatment strategy for these challenging lesions.

Thus, we propose the use of a true osteoligamentous three-column concept for all proximal ulna fractures to restore 360º stability adequately and anatomically to the elbow.”

2.

Materials and methods, Patient selection

2.1.

Is there informed consent from the patients for the use of their data, or is this established in the protocol approved by the ethics committee? It is necessary to make it clear or show that there is evidence of this.

2.2.

Is age and gender classification necessary? Is this included in the analysis?

2.3.

The methodology for obtaining the tomography (CT) is mentioned; however, the methodology used to analyze the data to obtain the information needs to be clarified. It is necessary to include it.

Thanks again. Regarding your first question, all patients signed an Informed Consent form. This information was added in the text.

Regarding your second question about age and gender classification, this is mentioned in the 2nd paragraph. As highlighted in the end of this paragraph, women were significantly older than men (p=0.007, Mann-Whitney test).

Finally, regarding your third question, using the axial, oblique coronal, and sagittal images, and 3D reconstruction, the proposed classification was applied using as markers the sublime tubercle as the medial column; the supinator crest as the lateral column; and the coronoid process of the ulna and the olecranon as the intermediate column of the ulna. As this is explained during the Material and Methods section, as you agree, we feel this don’t need further detailing.

Material and Methods, Patient selection, 1st paragraph, lines 108-110

“This study was approved by a human research ethics committee with protocol number 49409121.2.1001.5127. All patients signed an Informed Consent form.”

3.

Materials and methods, Classification

3.1.

Are the images in this section based on some reference, or are they your own? Because there is no support.

3.2.

It could be seen better and easier to read if the classification with the images is organized as columns or in a table (As a recommendation). The view must also be identified in each image of each figure

Responding your first question, all images are from the authors’ personal archive. The source is not mentioned, since the classification is nine and proposed in the study, and because it does not belong to any other author.

In regards of your second question, Table 2 shows affected column structures according to type in the new classification. The view was identified in each figure. Thanks for the suggestion.

Material and Methods, Classification, 1st paragraph, lines 142-145

“Figures 2 to 8 bring the classification with the affected structures in the three columns and illustrated always showing the medial view of the elbow on the left side and the lateral view of the elbow on the right side.”

4.

Materials and methods, Validation

How to guarantee that the position of the cases changes randomly throughout the process and does not repeat itself simultaneously?

Thanks for mentioning this aspect. The information was added in this section.

Material and Methods, Validation, lines 214-216

“Between the first and second rounds, all images were allocated and identified by the main author (PJL) and the positions of cases were randomly and manually changed by the principal researcher.”

5.

Results, Intra-rater agreement

Although a good agreement is shown between the 3 raters involved in the test, in order to carry out a validation process regarding this type of fracture, more opinions are required to allow a complete statistical approximation.

The validation criterion of the classification proposed in this study was not in the clinical setting. In this first phase of the classification development process, based on the theory of the 3 osteoligamentous columns of the elbow, the types were generated, and their validity was assessed to understand their degree of understanding and intra- and interobserver agreement. Our study showed good intra- and inter-rater agreement, attesting to the stability of the proposed classification among the raters, regardless of the level of experience of each one. The next phase of this study is to assess its usefulness in the clinical scenario, as stated in lines 317 to 319 of the Discussion.

Discussion, 6th paragraph, lines 317-319

“The next step for this consortium of investigators is to carry out a multicenter prospective cohort study soon to validate the new classification in a clinical-therapeutic setting.”

6.

Discussion

6.1.

The conclusion of a "very good" agreement can be risky, as mentioned. It should be kept clear that only 3 rates were considered.

6.2.

In general, the errors that could be associated with a lousy classification following this methodology, what implications do they have, or could they have for the patient?

Thanks again. Regarding your first comment, this limitation is clearly stated in the last paragraph of the Discussion section, where we write “First, the new classification system was assessed by three orthopedic surgeons with dif-ferent levels of expertise, which may have influenced our findings. However, we found good intra- and inter-rater agreement, which can be seen as a promising perspective for understanding and applying the proposed classification.”.

Finally, regarding the errors that may be associated with a poor classification according to the methodology used in the study, it is not possible to know or even understand the potential implications they have for the patient. As mentioned, this is phase 1 of the complete validation of our classification. However, the concept of three columns for the elbow joint is not new. The novelty of our classification is to gather all the osteoligamentous structures of the medial, intermediate (or middle) and lateral columns. Theoretically, there are no major implications, as we are using anatomical landmarks and functional structures for stabilization and normal range of motion of this joint.

Discussion, 6th paragraph, lines 325-329

“First, the new classification system was assessed by three orthopedic surgeons with different levels of expertise, which may have influenced our findings. However, we found good intra- and inter-rater agreement, which can be seen as a promising perspective for understanding and applying the proposed classification.”

For more details please see the revised version manuscript.

Round 2

Reviewer 1 Report

The author carefully replied to the reviewer's questions, and the paper was revised according to the reviewer's comments.